# Adapt the Face, Not the Voice: Asymmetric Fine-Tuning of Foundation Models for Cross-Modal Person Matching

## Abstract

Cross-modal person matching - associating a person's voice with their face - requires bridging speech and vision representations that share no direct physical correspondence. We investigate a simple approach: pairing frozen unimodal foundation models (WavLM-Large for speech, SigLIP ViT-B/16 for faces) with lightweight trainable projections into a shared embedding space. Our central finding is an informative asymmetry in the effectiveness of Low-Rank Adaptation (LoRA): adapting the face encoder yields substantial gains while adapting the voice encoder provides no benefit. We explain this asymmetry through layer-wise identity probing: WavLM already encodes strong speaker identity information (93.8% linear probe accuracy on 70 classes), while SigLIP's face identity representations are comparatively weak (79.5%), leaving substantially more room for task-specific adaptation. This gap widens on a larger evaluation: on 1,211-identity VoxCeleb1, WavLM maintains 90.5% probe accuracy while SigLIP drops to 58.1%. The asymmetric LoRA finding replicates across three datasets—MAV-Celeb (70 identities, per-identity split), VoxCeleb1 (1,211 identities, identity-disjoint split), and CN-Celeb-AV (689 identities, identity-disjoint, Chinese-language multi-genre)—and across evaluation protocols including verification, retrieval, and $N$-way matching. On MAV-Celeb, face-only LoRA achieves $16.6 \pm 0.4\%$ Equal Error Rate (mean $\pm$ std over 3 seeds) with only 1.33M trainable parameters (0.32% of the encoder total), compared to 19.9% for the prior best published result under a comparable (though not identical) evaluation protocol. Our results suggest a hypothesis for cross-modal adaptation: *selectively adapting the encoder whose pretraining is least aligned with the target task may be the key factor determining adaptation effectiveness.*

## 1 Introduction

Humans can associate faces with voices after brief exposure, leveraging subtle correlations in identity-related cues across modalities (Nagrani et al., 2018b; Kim et al., 2018). Replicating this ability computationally - cross-modal person matching - has applications in speaker diarization, media indexing, and multimodal person re-identification. The task is challenging because the statistical dependencies between facial appearance and vocal characteristics are indirect, mediated by shared physiological and demographic factors rather than direct physical correspondence.

The FAME challenge (Saeed et al., 2024) established a standardized benchmark for face-voice association on the MAV-Celeb dataset (Saeed et al., 2021). The current state of the art (MFV-KSD; Tao et al. 2024) achieves 19.9% Equal Error Rate (EER) through a complex three-stage pipeline using ECAPA-TDNN (Desplanques et al., 2020) and ResNet-18 encoders with keynote speaker diarization. The only foundation-model-based approach to date used ImageBind (Girdhar et al., 2023) with LoRA adapters, achieving 24.7% EER on the subsequent FAME 2026 challenge (Farhadipour et al., 2025)—suggesting that foundation models have untapped potential for this task.

We investigate a simpler approach: pair *best-in-class unimodal* foundation models - WavLM-Large (Chen et al., 2022) for speech and SigLIP ViT-B/16 (Zhai et al., 2023) for faces - with lightweight projection heads that map their frozen representations into a shared embedding space. Even this minimal setup achieves

$19.9 \pm 0.3\%$ EER with only 1.18M trainable parameters, matching the best published FAME challenge result of 19.9% EER (under a comparable but not identical evaluation protocol; see Section 4.2).

Our central finding concerns the effect of Low-Rank Adaptation (Hu et al., 2022) applied to the frozen encoders. In a systematic ablation, we discover a striking asymmetry: LoRA on the face encoder (SigLIP) reduces EER from $19.9 \pm 0.3\%$ to $16.6 \pm 0.4\%$, while LoRA on the voice encoder (WavLM) provides *no benefit whatsoever* ($19.8 \pm 0.2\%$ EER). This pattern - face adaptation is the critical ingredient - replicates on VoxCeleb1 (Nagrani et al., 2017), a larger dataset with an identity-disjoint evaluation protocol.

We provide a mechanistic explanation through layer-wise identity probing (Alain & Bengio, 2017). Training linear classifiers on intermediate representations reveals that WavLM encodes strong speaker identity from early layers (93.8% accuracy on MAV-Celeb, 90.5% on VoxCeleb1), while SigLIP's face identity encoding is substantially weaker (79.5% and 58.1%, respectively). The voice encoder already "knows" who is speaking; the face encoder does not reliably "know" whose face it sees. LoRA adaptation is effective where the identity information gap is largest—on the face encoder whose pretrained representations encode identity less strongly.

Our contributions are:

1. We demonstrate that frozen unimodal foundation models with lightweight projection heads (1.18M parameters, 0.29% of total) achieve strong cross-modal matching on MAV-Celeb ($16.6 \pm 0.4\%$ EER with face-only LoRA, 1.33M parameters), matching or surpassing published FAME challenge results.
2. We identify and explain a striking asymmetry: LoRA on the face encoder is highly effective while LoRA on the voice encoder provides no benefit, explained by a corresponding asymmetry in pretrained identity encoding revealed through layer-wise probing.
3. We replicate this finding on VoxCeleb1 and CN-Celeb-AV with identity-disjoint evaluation protocols spanning different demographics and languages, validate it with a second vision encoder (DINOv2), and test across multiple evaluation paradigms (*N*-way matching, cross-dataset transfer).

## 2 Related Work

**Face-voice association.** Nagrani et al. (2018b) first demonstrated that neural networks can learn to associate faces with voices, framing the problem as cross-modal verification and retrieval. Subsequent work explored shared embedding spaces (Nagrani et al., 2018a; Wen et al., 2019), hyperbolic alignment (Hannan et al., 2025), and cross-modal attention (Kim et al., 2018). The FAME challenge (Saeed et al., 2024) standardized evaluation on MAV-Celeb (Saeed et al., 2021), with MFV-KSD (Tao et al., 2024) achieving the current best EER of 19.9% through a multi-stage pipeline. All top-performing FAME 2024 methods use classic encoders (ECAPA-TDNN, ResNet-18); Farhadipour et al. (2025) represent the only foundation-model approach, using ImageBind with LoRA in the subsequent FAME 2026 challenge. Our work differs by using specialized unimodal foundation models rather than a single multimodal model, and by systematically studying where adaptation is needed.

**Foundation models for speech and vision.** Self-supervised speech models have achieved strong results across speech processing tasks. WavLM (Chen et al., 2022) and wav2vec 2.0 (Baevski et al., 2020) learn general speech representations through masked prediction, with WavLM achieving near-perfect speaker verification on the SUPERB benchmark (Yang et al., 2021). Layer-wise analyses show that self-supervised speech models encode acoustic, phonetic, and word-level information at different depths (Pasad et al., 2021). Probing for speaker-specific attributes reveals that speaker identity peaks in early-to-middle layers, though larger models can partially recover speaker information in deeper layers (Chiu et al., 2025). Vision transformers pretrained with contrastive objectives - CLIP (Radford et al., 2021), SigLIP (Zhai et al., 2023) - produce versatile visual representations but are not specifically trained for person-level discrimination, unlike face recognition models such as ArcFace (Deng et al., 2019). Self-supervised vision models like DINOv2 (Oquab et al., 2024) learn rich visual features through self-distillation without text supervision; we use DINOv2 as a second vision encoder to test the generality of our findings.

**Parameter-efficient adaptation.** LoRA (Hu et al., 2022) enables efficient adaptation of large pretrained models by injecting low-rank updates into attention weight matrices. While widely studied for language

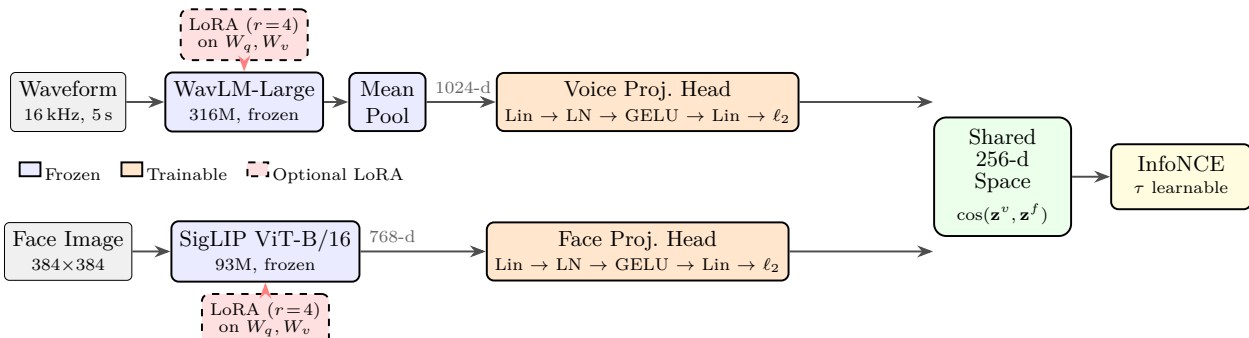

Figure 1: **Architecture overview.** Frozen WavLM-Large and SigLIP ViT-B/16 encoders produce utterance- and face-level representations. Trainable two-layer MLP projection heads (orange) map these to a shared 256-d unit hypersphere. Symmetric InfoNCE with learnable temperature $\tau$ trains the projections. Optional rank-4 LoRA adapters (dashed red) inject updates into encoder attention layers. Our key finding: only the face encoder LoRA (bottom) contributes meaningfully.

models, its application to cross-modal settings involving heterogeneous encoder pairs is less explored. Our finding that LoRA effectiveness is modality-dependent in cross-modal settings adds a new dimension to the understanding of when and where adaptation is needed.

**Probing neural representations.** Linear probing - training a linear classifier on frozen intermediate representations - is a standard diagnostic for understanding what information neural networks encode at each layer (Alain & Bengio, 2017; Conneau et al., 2018). Pasad et al. (2021) applied this methodology to wav2vec 2.0, analyzing acoustic, phonetic, and word-level properties across layers. Chiu et al. (2025) extended probing to speaker-specific attributes across 11 self-supervised models, finding that speaker identity generally concentrates in early-to-middle layers—though larger models can partially recover it in deeper layers. We extend this approach to a cross-modal setting, using probing to explain an observed asymmetry in adaptation effectiveness between speech and vision encoders.

## 3 Method

Our architecture follows the dual-encoder paradigm of CLIP (Radford et al., 2021), adapted for voice-face association. Two frozen foundation models encode each modality independently, and lightweight trainable projection heads map their representations into a shared embedding space where identity matching is performed via cosine similarity. Figure 1 illustrates the full pipeline.

### 3.1 Frozen Encoders

For voice, we use **WavLM-Large** (Chen et al., 2022), a 24-layer transformer (316M parameters) pretrained on 94K hours of speech via masked speech prediction and denoising. It produces 1024-dimensional frame-level representations, which we mean-pool over the time axis to obtain a single utterance embedding. WavLM achieves 0.617% EER on VoxCeleb1 speaker verification (Chen et al., 2022), indicating that its frozen representations encode strong speaker-discriminative features.

For faces, we use **SigLIP ViT-B/16** (Zhai et al., 2023; Dosovitskiy et al., 2021), a 12-layer vision transformer (93M parameters) pretrained on WebLI with a sigmoid contrastive loss for image-text matching. It produces a 768-dimensional pooled representation from $384 \times 384$ face images. Unlike face recognition models such as ArcFace (Deng et al., 2019), SigLIP was not trained for person-level discrimination - a distinction that proves critical for understanding our results.

Both encoders are frozen by default; all parameters have `requires_grad=False`.

### 3.2 Projection Heads

Each modality has a dedicated two-layer MLP projection head:

$$\mathbf{z} = \ell_2\text{-norm}\big(\mathbf{W}_2 \cdot \text{GELU}(\text{LN}(\mathbf{W}_1\mathbf{h} + \mathbf{b}_1)) + \mathbf{b}_2\big), \tag{1}$$

where $\mathbf{h} \in \mathbb{R}^{d_{\text{enc}}}$ is the frozen encoder output, $\mathbf{W}_1 \in \mathbb{R}^{512 \times d_{\text{enc}}}$, $\mathbf{W}_2 \in \mathbb{R}^{256 \times 512}$, LN denotes LayerNorm, and $\ell_2$-normalization maps the output to the unit hypersphere $\mathcal{S}^{255}$. The voice head has $\sim$656K parameters; the face head has $\sim$524K parameters, totaling 1.18M.

### 3.3 Training Objective

We use symmetric InfoNCE (van den Oord et al., 2018) with a learnable log-temperature $\tau$ (initialized to $\log(1/0.07)$):

$$\mathcal{L} = \frac{1}{2}\big(\mathcal{L}_{\text{v}\to\text{f}} + \mathcal{L}_{\text{f}\to\text{v}}\big), \quad \mathcal{L}_{\text{v}\to\text{f}} = -\frac{1}{B}\sum_{i=1}^{B} \log \frac{\exp(\mathbf{z}_i^v \cdot \mathbf{z}_i^f/\tau)}{\sum_{j=1}^{B} \exp(\mathbf{z}_i^v \cdot \mathbf{z}_j^f/\tau)}, \tag{2}$$

where $B$ is the batch size and $(\mathbf{z}_i^v, \mathbf{z}_i^f)$ are voice and face embeddings of the same identity.

### 3.4 LoRA Adaptation

To probe whether limited encoder fine-tuning improves alignment, we optionally inject Low-Rank Adaptation (LoRA; Hu et al. 2022) modules into the query and value projections ($W_q$, $W_v$) of each transformer's self-attention layers. We use rank $r = 4$ with $\alpha = 8$ and no dropout, adding 147K parameters for SigLIP (12 layers $\times$ 2 matrices) and 393K for WavLM (24 layers $\times$ 2 matrices). When LoRA is active, gradients flow through the adapter weights while all other encoder parameters remain frozen.

We evaluate four configurations:

- **Proj-only**: Only projection heads are trained (1.18M params).
- **+LoRA voice**: Projection heads + LoRA on WavLM (1.58M params).
- **+LoRA face**: Projection heads + LoRA on SigLIP (1.33M params).
- **+LoRA both**: Projection heads + LoRA on both encoders (1.72M params).

### 3.5 Optimization

We train with AdamW (weight decay $10^{-2}$) using OneCycleLR scheduling with 5% linear warmup and cosine annealing. The learning rate is $10^{-3}$ for projection-only and $5 \times 10^{-4}$ when LoRA is active. We use mixed-precision training (bfloat16) with gradient clipping at norm 1.0. Batch sizes are 64 for projection-only and 32 with LoRA.

## 4 Experiments

### 4.1 Experimental Setup

**Datasets.** We evaluate on three datasets spanning different scales, demographics, and evaluation protocols:

**MAV-Celeb v1** (Saeed et al., 2021) is the standard benchmark for the FAME challenge (Saeed et al., 2024), containing multilingual celebrity videos. We use 70 identities with an 80/20 per-identity train/test split following prior work. This split means the same identities appear in both training and testing, with disjoint samples. Specifically, for each of the 70 identities, we randomly assign 80% of audio clips and 80% of face images to the training set and the remaining 20% to the test set (with a fixed random seed for reproducibility). This per-identity split differs from the official FAME challenge protocol (Saeed et al., 2024) in several ways. The FAME challenge uses MAV-Celeb v1 with its own predefined train/validation/test partitions, which include identity overlap across splits but with challenge-specific data selection, preprocessing, and evaluation scripts. Our 70-identity subset with an 80/20 per-identity split was chosen to follow the

general evaluation setup used in related work on this dataset, but is not identical to the official FAME protocol. As a consequence, the comparison to the published FAME SOTA of 19.9% EER (Tao et al., 2024) is *approximate*: while both evaluations measure EER on MAV-Celeb v1, differences in the exact data partition, the number of test samples, and the pair construction procedure mean that EER values are not directly comparable. We report the FAME result as a reference point for the general difficulty of the task rather than as a controlled comparison.

**VoxCeleb1** (Nagrani et al., 2017) contains over 153K utterances from 1,251 celebrities. We extract face images from the original YouTube videos and construct an identity-disjoint split: 1,211 identities for training and 40 held-out identities for testing (derived from the verification test pairs). This is a substantially harder evaluation: the model must generalize to *entirely unseen* individuals. We note that VoxCeleb1's official `iden_split.txt` file assigns individual *utterances* (not identities) to train/validation/test splits, so every identity appears in all three partitions. This makes it unsuitable for identity-disjoint evaluation. Instead, we derive our 40 test identities from `veri_test.txt`, the official VoxCeleb1-E verification protocol file, which lists trial pairs involving exactly 40 distinct speaker IDs. All remaining 1,211 identities (with both audio and face data available) are used for training. For verification evaluation, we construct 10,000 balanced cross-modal pairs from the 40 test identities: each pair consists of one audio clip and one face image, drawn from the same identity (positive, ~50%) or different identities (negative, ~50%), sampled with a fixed random seed. EER is computed from cosine similarity scores over these pairs. For retrieval evaluation, we compute per-identity mean embeddings (averaged over all test samples for each identity, then re-normalized to unit norm) and report Rank-$K$ accuracy as the fraction of queries for which the correct identity appears among the $K$ nearest neighbors, averaged over voice-to-face and face-to-voice directions. The retrieval evaluation thus operates over all 40 test identities, where chance-level Rank-1 accuracy is 2.5%.

**CN-Celeb-AV** (Li et al., 2023) is a multi-genre audio-visual dataset of 1,136 Chinese celebrities, covering 11 genres (interview, entertainment, singing, drama, etc.) collected from Bilibili. We use the official identity-disjoint splits: 689 identities for training (`dev`, 93,973 utterances with paired face frames) and 197 held-out identities for evaluation (`eval_f`, with 70,772 cross-modal verification trial pairs). CN-Celeb-AV provides demographic diversity (Chinese-language media, compared to the multilingual MAV-Celeb and English-dominant VoxCeleb1) and genre diversity (singing and drama segments alongside interviews), making it a challenging complement to our other evaluations.

**Metrics.** We report **Equal Error Rate** (EER; lower is better), the primary metric of the FAME challenge; and **Rank-$K$** retrieval accuracy for $K \in \{1, 5, 10\}$ (higher is better), computed as the average of voice-to-face and face-to-voice retrieval using per-identity mean embeddings.

**$N$-way matching protocol.** Following Nagrani et al. (2018b), we evaluate $N$-way matching: given a voice query, select the correct face from $N$ candidates (1 target + $N-1$ distractors). We test $N \in \{2, 5, 10, 20\}$ plus the full gallery size ($N=70$ for MAV-Celeb, $N=40$ for VoxCeleb1). Small $N$ tests basic discriminability (e.g., $N=2$ is a forced-choice pair), while $N$ equal to the full test set measures open-set identification difficulty; intermediate values trace the degradation curve between these extremes. We report accuracy averaged over 500 trials per test identity with random distractor sampling.

**Epoch selection.** To avoid test-set peeking, we report results at a fixed training epoch for all models: the final checkpoint (epoch 30 for MAV-Celeb and CN-Celeb-AV, epoch 10 for VoxCeleb1). Training dynamics across all epochs are provided in Appendix C.

## 4.2 MAV-Celeb Ablation

Table 1 presents our ablation study on MAV-Celeb.

**Frozen encoders are strong bases.** With zero training, cross-modal matching is at chance (50.4% EER). However, training only the lightweight projection heads achieves $19.9 \pm 0.3\%$ EER, essentially matching the best published FAME result of 19.9%. Note that the FAME evaluation protocol differs from ours (they use the official FAME splits, while we use a 70-identity subset with 80/20 per-identity split), so this comparison is

Table 1: **Ablation study on MAV-Celeb v1** (70 identities, 80/20 per-identity split). Trainable parameters shown as count and percentage of total (409M). Results at epoch 30, reported as mean ± std over 3 random seeds. Best result per column in **bold**. FAME SOTA shown for approximate reference; note that MFV-KSD used a different evaluation protocol (see text).

| Configuration | Trainable | Rank-1 | Rank-5 | Rank-10 | EER↓ |
|---|---|---|---|---|---|
| *Baselines* | | | | | |
| Zero-shot (no training) | 0 | 0.7% | 7.9% | 15.0% | 50.4% |
| *FAME SOTA (MFV-KSD)* | — | — | — | — | *19.9%* |
| *Our models (mean ± std, n=3 seeds)* | | | | | |
| Proj-only | 1.18M (0.29%) | 69.8±1.2% | 92.6±0.7% | 96.4±0.0% | 19.9±0.3% |
| + LoRA voice | 1.58M (0.38%) | 71.4±1.0% | 90.2±1.2% | 96.7±1.2% | 19.8±0.2% |
| + LoRA face | 1.33M (0.32%) | **80.7±0.6%** | **94.3±0.6%** | **98.6±0.6%** | **16.6±0.4%** |
| + LoRA both | 1.72M (0.42%) | 80.3±2.4% | 93.8±1.2% | 97.6±0.9% | 16.9±0.7% |

approximate. Regardless, the result demonstrates that WavLM and SigLIP encode sufficient identity-relevant information in their frozen representations to enable strong cross-modal matching with minimal adaptation.

**Face adaptation matters; voice adaptation does not.** The key result is the asymmetry between voice and face adaptation. Adding LoRA to WavLM ("+LoRA voice") yields $19.8 \pm 0.2\%$ EER - *no improvement* over projection-only ($19.9 \pm 0.3\%$). In contrast, adding LoRA to SigLIP ("+LoRA face") reduces EER to $16.6 \pm 0.4\%$. Retrieval metrics show a consistent pattern: Rank-1 accuracy improves from $69.8 \pm 1.2\%$ to $80.7 \pm 0.6\%$ with face LoRA, while voice LoRA provides negligible change ($71.4 \pm 1.0\%$). The non-overlapping error bars confirm that face adaptation is a statistically robust improvement.

**Combining both adaptations.** Adding LoRA to both encoders yields $16.9 \pm 0.7\%$ EER, comparable to but slightly worse than face-only LoRA ($16.6 \pm 0.4\%$). Rank-1 accuracy is similar ($80.3 \pm 2.4\%$ vs. $80.7 \pm 0.6\%$), though with notably higher variance, suggesting that the additional voice parameters introduce instability without benefit. In this setting, face LoRA alone accounts for the full adaptation benefit; voice LoRA adds no value.

**Parameter count confound.** One might argue that LoRA-face (1.33M parameters) outperforms LoRA-voice (1.58M) simply because it adapts different layers, not because face adaptation is inherently more useful. To rule out a parameter-count confound, we train a rank-1 LoRA-voice variant with ~1.28M parameters - closely matching LoRA-face's 1.33M. This model achieves 20.1% EER and 68.6% Rank-1, *worse* than projection-only ($19.9 \pm 0.3\%$ / $69.8 \pm 1.2\%$) and far below LoRA-face ($16.6 \pm 0.4\%$ / $80.7 \pm 0.6\%$). The improvement from face LoRA is thus attributable to *where* adaptation is applied, not how many parameters are added.

Table 2 contextualizes our results against prior work on the FAME benchmark.

The ImageBind+LoRA approach (Farhadipour et al., 2025), evaluated under the FAME 2026 protocol, is the most directly comparable foundation-model baseline. Despite ImageBind's 1.2B parameters, it achieves only 24.7% EER. We attribute our superior performance to the use of *best-in-class unimodal* encoders: WavLM-Large specifically excels at speaker-discriminative representations, while SigLIP provides strong visual features. This supports the principle that specialized unimodal foundation models, bridged by lightweight alignment layers, can outperform larger multimodal models on targeted cross-modal tasks.

### 4.3 Why Face Adaptation Matters: Identity Probing Analysis

The asymmetric LoRA result raises a natural question: *why* does the face encoder benefit from adaptation while the voice encoder does not? We hypothesize that this reflects a corresponding asymmetry in how much identity information each encoder captures in its pretrained representations.

Table 2: **Comparison with prior work on MAV-Celeb.** Baseline EER values from published results under FAME challenge protocols. ImageBind+LoRA is from FAME 2026; all other baselines are from FAME 2024. Our results use a 70-identity subset with 80/20 per-identity split, which differs from the official protocol; this comparison is therefore approximate.

| Method | Encoders | Approach | EER↓ |
|---|---|---|---|
| FOP baseline | ResNet-18 + TDNN | Fusion + orth. proj. | 33.4% |
| ImageBind+LoRA | ImageBind (1.2B) | LoRA + contrastive | 24.7% |
| SCC | Transformer enc. | Contrastive + cluster | ∼20.5% |
| MFV-KSD | ECAPA-TDNN + ResNet-18 | 3-stage + diarization | 19.9% |
| Ours (proj-only) | WavLM-L + SigLIP | Proj. heads (1.18M) | 19.9±0.3% |
| **Ours (+LoRA face)** | WavLM-L + SigLIP | + LoRA face (1.33M) | **16.6±0.4%** |
| Ours (+LoRA both) | WavLM-L + SigLIP | + LoRA both (1.72M) | 16.9±0.7% |

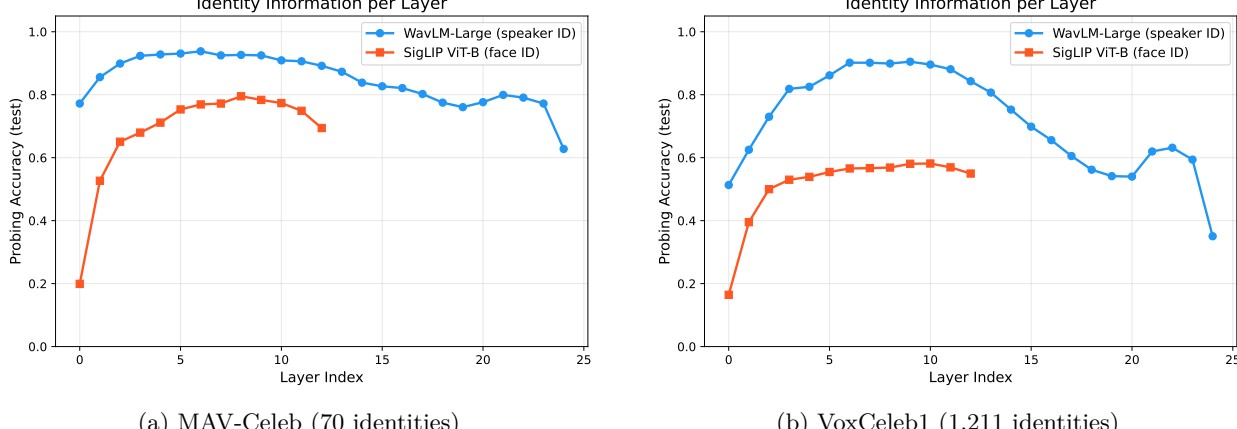

(a) MAV-Celeb (70 identities)  (b) VoxCeleb1 (1,211 identities)

Figure 2: **Layer-wise identity probing accuracy** for WavLM-Large (speaker ID, blue) and SigLIP ViT-B/16 (face ID, orange). Linear classifiers are trained per layer on frozen representations. WavLM encodes strong speaker identity across all three datasets, while SigLIP's face identity encoding is substantially weaker—especially on the larger-scale evaluations. This gap explains why face LoRA is effective: SigLIP needs task-specific adaptation to better encode facial identity.

To test this, we conduct layer-wise identity probing (Alain & Bengio, 2017) on both encoders. For each transformer layer, we extract representations for all samples in the training set (mean-pooled over time for WavLM, CLS token for SigLIP), train a logistic regression classifier to predict speaker/face identity, and evaluate on held-out samples. This measures how much identity-discriminative information is linearly accessible at each layer - a standard diagnostic for what neural networks "know" about a given attribute (Conneau et al., 2018).

Figure 2 shows the results on MAV-Celeb and VoxCeleb1 (CN-Celeb-AV probing results are presented in Section 4.5). The findings are striking:

**WavLM encodes strong speaker identity.** On MAV-Celeb (70 identities), WavLM achieves 93.8% probing accuracy at its best layer (layer 6), with 100% training accuracy at nearly all layers. On VoxCeleb1 (1,211 identities), accuracy remains high at 90.5% (layer 9) despite the 17× increase in identity count. Speaker identity information is concentrated in early-to-middle layers (4–9), consistent with prior probing analyses of speaker-specific attributes in self-supervised speech models (Chiu et al., 2025). Later layers (18–24) show declining identity accuracy, likely reflecting a shift toward linguistic content representations.

Table 3: **Ablation study on VoxCeleb1** (1,211 train / 40 test identities, identity-disjoint split). Results at the final training epoch (epoch 10). Best result per column in **bold**.

| Configuration | Trainable | Rank-1 | Rank-5 | Rank-10 | EER↓ |
|---|---|---|---|---|---|
| Zero-shot | 0 | 1.2% | 11.2% | 27.5% | 50.2% |
| Proj-only | 1.18M | 6.2% | 35.0% | 57.5% | 41.6% |
| + LoRA voice | 1.58M | 7.5% | 43.8% | 63.7% | 39.8% |
| + LoRA face | 1.33M | **15.0%** | 46.2% | 62.5% | **37.5%** |
| + LoRA both | 1.72M | **15.0%** | **51.2%** | **71.2%** | 38.1% |

**SigLIP encodes face identity much less strongly.** On MAV-Celeb, SigLIP peaks at 79.5% (layer 8) - 14.3 percentage points below WavLM. On VoxCeleb1, the gap widens dramatically: SigLIP achieves only 58.1% (layer 9–10), a 32.4pp deficit. The training accuracy never reaches 100% on VoxCeleb1 (peaking at 84.7%), indicating that SigLIP's representations cannot even perfectly separate training identities.

**The identity gap predicts adaptation effectiveness.** The probing results provide a clear explanation for the asymmetric LoRA finding. WavLM's representations already encode near-ceiling speaker identity; there is little room for LoRA to add information, and the additional parameters risk overfitting. SigLIP's representations leave substantial identity information "on the table" - LoRA adaptation fills this gap by steering the face encoder toward identity-discriminative features that the projection heads can then align with the voice representations.

The gap is consistent across all three datasets (14.3pp on MAV-Celeb, 32.4pp on VoxCeleb1, 15.6pp on CN-Celeb-AV) and correctly predicts where LoRA helps in each case.

### 4.4 VoxCeleb1 Ablation

To test whether the asymmetric LoRA finding generalizes beyond MAV-Celeb, we replicate the full ablation on VoxCeleb1 with an identity-disjoint evaluation protocol. This is a substantially harder setting: the model must generalize to 40 entirely unseen identities from a training set of 1,211 identities.

Table 3 shows the results. The asymmetric pattern replicates clearly:

**Face LoRA is the best single intervention.** LoRA on SigLIP alone achieves the best EER (37.5%) and matches the best Rank-1 accuracy (15.0%) - while using fewer parameters than LoRA-both. On MAV-Celeb, LoRA-both achieved a slightly better EER than LoRA-face; here, LoRA-face is competitive or better, further supporting the asymmetry.

**Voice LoRA provides minimal benefit.** LoRA on WavLM produces 39.8% EER and 7.5% Rank-1, barely improving over the projection-only baseline (41.6% EER, 6.2% Rank-1), replicating the MAV-Celeb observation that voice adaptation is largely unnecessary.

**Voice LoRA hurts when combined with face LoRA.** On VoxCeleb1, adding voice LoRA to face LoRA *degrades* EER: it rises from 37.5% to 38.1%. This suggests that voice LoRA adapts WavLM to training-set-specific speaker characteristics that do not transfer to unseen identities.

The overall EER values on VoxCeleb1 (37–42%) are substantially higher than on MAV-Celeb (14–19%). This gap reflects the harder evaluation protocol (identity-disjoint vs. per-identity split), the larger identity space (1,211 vs. 70 training identities), and lower face quality (YouTube-extracted frames vs. curated celebrity images). Importantly, the *relative ordering* of methods is consistent across datasets, confirming the robustness of the asymmetric adaptation finding.

Table 4: **Ablation study on CN-Celeb-AV** (689 train / 197 test identities, identity-disjoint split). Results at epoch 30. Best result per column in **bold**.

| Configuration | Trainable | Rank-1 | EER↓ |
|---|---|---|---|
| Proj-only | 1.18M | 37.4% | 40.3% |
| + LoRA voice | 1.58M | 35.8% | 41.3% |
| **+ LoRA face** | 1.33M | **47.7%** | **37.2%** |
| + LoRA both | 1.72M | 48.0% | 38.4% |

### 4.5 CN-Celeb-AV Ablation

To further test the generality of the asymmetric LoRA finding across demographics and languages, we replicate the ablation on CN-Celeb-AV (Li et al., 2023), a multi-genre audio-visual dataset of Chinese celebrities. This dataset differs from MAV-Celeb and VoxCeleb1 in several important respects: it features Chinese-language media (vs. multilingual and English), covers 11 diverse genres (interviews, singing, drama, entertainment, etc.), and provides official identity-disjoint evaluation splits (689 train / 197 test identities).

Table 4 shows the results. The asymmetric pattern replicates clearly on this third dataset:

**Face LoRA is again the key intervention.** LoRA on SigLIP achieves the best EER (37.2%) and the best Rank-1 accuracy (47.7%), improving over the projection-only baseline by 3.1 percentage points in EER and 10.3 points in Rank-1.

**Voice LoRA again provides no benefit.** LoRA on WavLM yields 41.3% EER—slightly *worse* than projection-only (40.3%)—replicating the pattern observed on MAV-Celeb and VoxCeleb1. Rank-1 accuracy (35.8%) is also below projection-only (37.4%), confirming that voice adaptation is consistently unhelpful.

**Identity probing confirms the explanation.** Layer-wise probing on CN-Celeb-AV (Figure 3) reveals the same identity encoding asymmetry: WavLM achieves 71.3% speaker identity probing accuracy (layer 6), while SigLIP reaches only 55.7% face identity accuracy (layer 9)—a 15.6 percentage point gap. This is consistent with the gaps observed on MAV-Celeb (14.3pp) and VoxCeleb1 (32.4pp), and correctly predicts that face adaptation will be effective while voice adaptation will not.

The replication on CN-Celeb-AV—with its distinct language, demographics, genre diversity, and identity-disjoint protocol—provides strong evidence that the asymmetric adaptation finding is not an artifact of a specific dataset or evaluation setup.

### 4.6 $N$-way Matching

We evaluate all models using the $N$-way matching protocol, which tests identity discrimination at varying gallery sizes. Table 5 reports results on MAV-Celeb and VoxCeleb1.

Several patterns emerge consistently across datasets and gallery sizes:

**Face LoRA is the key ingredient.** On MAV-Celeb, LoRA-face achieves the highest accuracy at all $N$ values (99.1% at $N=2$, 80.0% at $N=70$), outperforming LoRA-both (99.0% and 78.6%, respectively). On VoxCeleb1, LoRA-both is nominally best at small $N$ (81.2% at $N=2$), but LoRA-face matches or exceeds it at large $N$ (25.4% vs. 25.2% at $N=20$; tied at 15.0% for $N=40$). Crucially, on all datasets the gap between LoRA-face and LoRA-voice is far larger than between LoRA-face and LoRA-both, confirming that face adaptation is the dominant factor.

**Voice LoRA provides marginal benefit at best.** On MAV-Celeb, LoRA-voice is *worse* than projection-only at all $N$ values (e.g., 83.3% vs. 84.0% at $N=20$; 70.7% vs. 71.4% at $N=70$). On VoxCeleb1, LoRA-voice modestly improves over projection-only (e.g., 48.2% vs. 43.6% at $N=5$) but trails LoRA-face by a wide margin.

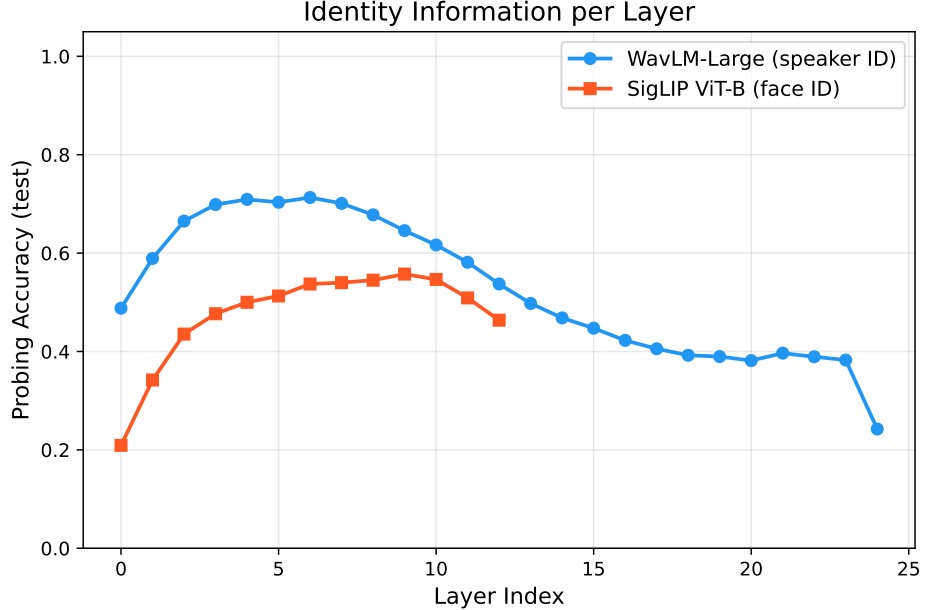

Figure 3: **Layer-wise identity probing on CN-Celeb-AV** (689 identities). WavLM (blue) peaks at 71.3% speaker identity accuracy (layer 6), while SigLIP (orange) peaks at 55.7% face identity accuracy (layer 9). The 15.6pp gap correctly predicts the asymmetric LoRA result on this dataset.

Table 5: $N$-**way matching accuracy** (bidirectional average of voice→face and face→voice, 500 trials per identity, higher is better). Chance level is $1/N$. Best trained model per column in **bold**.

| | MAV-Celeb (70 test IDs) | | | | |
|---|---|---|---|---|---|
| **Model** | $N=2$ | $N=5$ | $N=10$ | $N=20$ | $N=70$ |
| Random chance | 50.0% | 20.0% | 10.0% | 5.0% | 1.4% |
| Zero-shot | 49.2% | 19.7% | 9.8% | 5.0% | 1.4% |
| Proj-only | 98.4% | 94.5% | 89.9% | 84.0% | 71.4% |
| + LoRA voice | 98.4% | 94.2% | 89.4% | 83.3% | 70.7% |
| **+ LoRA face** | **99.1%** | **96.7%** | **93.7%** | **89.5%** | **80.0%** |
| + LoRA both | 99.0% | 96.3% | 93.2% | 88.7% | 78.6% |
| | VoxCeleb1 (40 test IDs, identity-disjoint) | | | | |
| **Model** | $N=2$ | $N=5$ | $N=10$ | $N=20$ | $N=40$ |
| Random chance | 50.0% | 20.0% | 10.0% | 5.0% | 2.5% |
| Zero-shot | 51.7% | 21.5% | 10.5% | 5.0% | 2.5% |
| Proj-only | 73.0% | 43.6% | 26.8% | 14.3% | 6.2% |
| + LoRA voice | 77.1% | 48.2% | 30.1% | 15.8% | 7.5% |
| + LoRA face | 78.4% | 51.7% | 37.0% | **25.4%** | **15.0%** |
| **+ LoRA both** | **81.2%** | **56.5%** | **39.9%** | 25.2% | **15.0%** |

**Scaling behavior.** On MAV-Celeb, even the projection-only model maintains strong accuracy (71.4% at $N=70$, where chance is 1.4%). On VoxCeleb1, performance degrades more steeply, reaching 6.2% for projection-only at $N=40$ (chance 2.5%) - still above chance, but indicating that the identity-disjoint protocol is substantially more challenging.

Table 6: **Cross-dataset transfer evaluation.** Models trained on one dataset are evaluated on the other's test set. In-domain results shown for reference.

| | Eval on VoxCeleb1 | | Eval on MAV-Celeb | |
|---|---|---|---|---|
| **Training source** | Rank-1 | EER | Rank-1 | EER |
| Zero-shot (no training) | 1.2% | 50.2% | 0.7% | 50.4% |
| *MAV-Celeb-trained models → VoxCeleb1 test* | | | | |
|   + LoRA face | 6.2% | 44.2% | — | — |
|   + LoRA both | 7.5% | 44.7% | — | — |
| *VoxCeleb1-trained models → MAV-Celeb test* | | | | |
|   + LoRA face | — | — | 3.6% | 43.2% |
|   + LoRA both | — | — | 6.4% | 42.0% |
|   + LoRA voice | — | — | 4.3% | 42.3% |
| *In-domain (for reference)* | | | | |
|   MAV-Celeb +LoRA face | — | — | 80.7% | 16.6% |
|   VoxCeleb1 +LoRA face | 15.0% | 37.5% | — | — |

## 4.7 Cross-Dataset Transfer

To assess whether the learned cross-modal alignment transfers across datasets and demographic distributions, we evaluate models trained on one dataset on the test set of the other. Table 6 presents the results.

**Cross-modal alignment transfers partially.** All cross-dataset models improve over the zero-shot baseline in EER (e.g., 44.2% vs. 50.2% for MAV→VC1), confirming that the learned voice-face mapping captures some domain-general structure. However, cross-dataset performance is far below in-domain performance (e.g., 6.2% vs. 15.0% Rank-1 for LoRA-face on VoxCeleb1), indicating that the alignment is largely dataset-specific.

**Both directions show similar degradation patterns.** Transferring from MAV-Celeb to VoxCeleb1, LoRA-face achieves 6.2% Rank-1 / 44.2% EER, while LoRA-both achieves 7.5% / 44.7%. In the reverse direction (VoxCeleb1 to MAV-Celeb), LoRA-both leads with 6.4% Rank-1 / 42.0% EER, versus 3.6% / 43.2% for LoRA-face. The EER values in both directions cluster tightly (42–45%), far from the zero-shot 50% baseline but also far from in-domain performance, suggesting that while the models learn some transferable cross-modal structure, much of the alignment is dataset-specific.

**LoRA-voice transfers comparably.** When transferring from VoxCeleb1 to MAV-Celeb, LoRA-voice (4.3% Rank-1, 42.3% EER) performs comparably to LoRA-face (3.6%, 43.2%), despite being much weaker in-domain. This suggests that in the cross-dataset regime, all methods are limited by the domain gap rather than by adaptation quality.

## 4.8 Generalization Across Vision Encoders

To test whether the asymmetric LoRA finding depends on the specific choice of vision encoder, we replace SigLIP with DINOv2 ViT-B/14 (Oquab et al., 2024) - a self-supervised vision model pretrained with a different objective (self-distillation vs. image-text contrastive learning). DINOv2 uses the same ViT-B architecture (768-d, 12 layers, 12 heads) but with combined Q/K/V attention projections, yielding exactly the same number of LoRA parameters as SigLIP (147K) for a fair comparison. Figure 4 illustrates the DINOv2 variant architecture.

We first probe DINOv2's frozen representations for face identity and find that it achieves **96.8%** accuracy on MAV-Celeb - substantially *higher* than SigLIP's 79.5%, and even exceeding WavLM's 93.8% speaker ID accuracy. DINOv2's self-supervised pretraining, optimized to produce robust visual features without text supervision, incidentally produces highly identity-discriminative face representations.

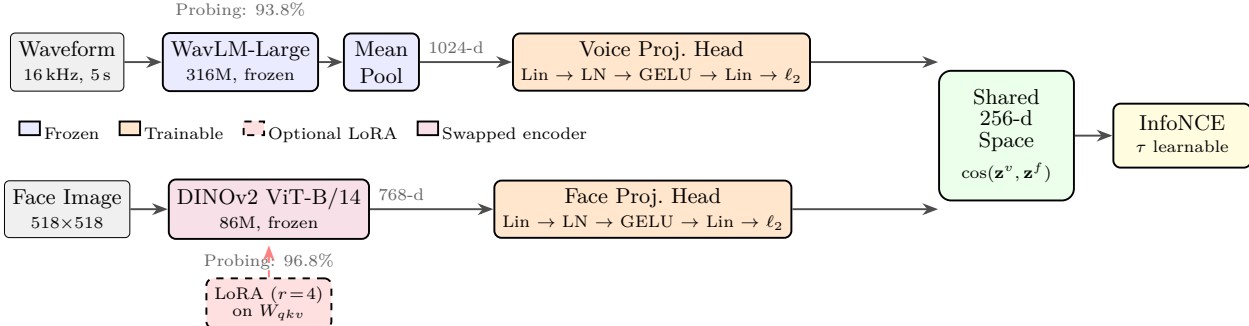

Figure 4: **DINOv2 variant architecture.** DINOv2 ViT-B/14 (purple) replaces SigLIP as the face encoder (cf. Figure 1). DINOv2's self-supervised pretraining yields 96.8% identity probing accuracy - higher than SigLIP's 79.5% and even WavLM's 93.8%. With identity already well-encoded, LoRA adaptation on DINOv2 provides no benefit (see Table 7). Note the combined $W_{qkv}$ projection in DINOv2 vs. separate $W_q, W_v$ in SigLIP.

Table 7: Effect of LoRA-face adaptation depends on encoder's identity probing accuracy. SigLIP (79.5% probing) benefits substantially from LoRA, while DINOv2 (96.8% probing) and ArcFace (dedicated face recognition) do not. All results on MAV-Celeb; SigLIP values are mean±std over 3 seeds.

| Face Encoder | Adaptation | Probing | EER $\downarrow$ | Rank-1 $\uparrow$ |
|---|---|---|---|---|
| SigLIP ViT-B/16 | Proj-only | 79.5% | 19.9±0.3% | 69.8±1.2% |
| SigLIP ViT-B/16 | + LoRA face | 79.5% | **16.6±0.4%** | **80.7±0.6%** |
| DINOv2 ViT-B/14 | Proj-only | 96.8% | 19.5% | 79.3% |
| DINOv2 ViT-B/14 | + LoRA face | 96.8% | 19.5% | 75.7% |
| ArcFace (IResNet) | Proj-only | — | 19.1% | 82.9% |

The probing-adaptation prediction is clear: since DINOv2 already encodes face identity at near-ceiling levels, LoRA adaptation should provide *less* benefit than it does for SigLIP. We test this prediction by training projection-only and LoRA-face variants on MAV-Celeb under the same protocol.

The results confirm the prediction. DINOv2 with projection-only achieves 19.5% EER and 79.3% Rank-1 - comparable EER to SigLIP projection-only ($19.9 \pm 0.3\%$) but with substantially higher Rank-1 (79.3% vs. $69.8 \pm 1.2\%$), consistent with DINOv2's stronger identity encoding. Critically, adding LoRA to DINOv2 provides *no benefit*: DINOv2 + LoRA-face achieves 19.5% EER and 75.7% Rank-1 - identical EER and slightly *worse* retrieval than projection-only. This contrasts sharply with SigLIP, where LoRA-face reduces EER by 3.3 percentage points (from $19.9 \pm 0.3\%$ to $16.6 \pm 0.4\%$) and improves Rank-1 by 10.9 points.

Table 7 summarizes the comparison, including a dedicated ArcFace face recognition model (discussed in Section 5) that further validates this pattern. These results demonstrate that the identity gap between encoders—not a specific architectural choice—determines the benefit of adaptation.

## 5    Discussion

**A hypothesis for cross-modal adaptation.**    Our results suggest a hypothesis: when bridging two frozen foundation models for cross-modal matching, *adaptation is most effective—and may only be needed—for the encoder whose pretraining is least aligned with the target task.* In our setting, WavLM - pretrained with objectives that capture speaker-discriminative features - already provides strong identity representations, while SigLIP - pretrained for general visual-semantic similarity - does not discriminate facial identity well. LoRA adaptation is effective precisely and only where this gap exists. Our experiments validate both directions of this prediction. SigLIP's low identity probing accuracy (79.5%) predicts effective adaptation, and indeed

**Face Encoder Comparison on MAV-Celeb**

Figure 5: **Face encoder comparison on MAV-Celeb.** All three encoders achieve similar EER with projection heads alone ($\sim$19–20%). Only SigLIP—the encoder with the weakest identity probing (79.5%)—benefits from LoRA adaptation (16.6% EER). ArcFace achieves the highest Rank-1 (82.9%) without any adaptation, consistent with its dedicated face recognition pretraining.

LoRA-face reduces EER from $19.9 \pm 0.3\%$ to $16.6 \pm 0.4\%$. Conversely, DINOv2's high probing accuracy (96.8%) predicts minimal benefit, and indeed LoRA-face provides no improvement (19.5% EER with or without LoRA).

If the hypothesis holds more broadly, identity probing could serve as a practical diagnostic: before committing to expensive adaptation experiments, probe each encoder's pretrained representations for the target attribute. The encoder with lower probing accuracy is the candidate for adaptation.

**Why not a dedicated face encoder?** One might ask why we do not simply use a face recognition model (e.g., ArcFace; Deng et al. 2019) instead of SigLIP. To test this directly, we replace SigLIP with a frozen ArcFace encoder (IResNet trained with angular margin loss on WebFace600K) and train projection heads only on MAV-Celeb. ArcFace proj-only achieves 19.1% EER and 82.9% Rank-1 (Table 7), confirming that a dedicated face recognition model performs well with alignment projections alone—no adaptation needed. Notably, ArcFace's Rank-1 (82.9%) exceeds even LoRA-adapted SigLIP ($80.7 \pm 0.6\%$), consistent with its superior identity encoding. However, ArcFace uses a CNN architecture (IResNet) that does not support transformer-based LoRA, making it less flexible for studying adaptation dynamics. We note that ArcFace requires tightly aligned 112×112 face crops; on VoxCeleb1, where faces were extracted via automated pipelines without ArcFace-specific alignment, performance degrades substantially (43.3% EER), whereas general-purpose vision encoders (SigLIP, DINOv2) are robust to such input variation. The DINOv2 result further reinforces this pattern: at 96.8% face identity probing accuracy, DINOv2 also shows no benefit from LoRA (19.5% EER with or without adaptation). Together, these results confirm that the benefit of adaptation is determined by the encoder's pretrained identity encoding, not by architectural choice. Figure 5 summarizes the comparison across all face encoders.

**Limitations.** Our evaluation has several limitations. First, the MAV-Celeb evaluation uses a per-identity split where the same identities appear in training and testing, which is a transductive setting that likely yields lower EER than an identity-disjoint protocol would. Our comparison to the FAME SOTA is therefore approximate: while the numerical values suggest competitive or superior performance, confirming this would require evaluation under the exact FAME protocol with its official splits and evaluation scripts. Second, VoxCeleb1 face images were extracted via automated pipelines and are lower quality than those in MAV-

Celeb, which affects absolute performance. Third, while we evaluate on three datasets spanning different demographics and languages, all three involve celebrity media; evaluation on non-celebrity or conversational data would further strengthen the generality of our claims. Finally, our probing experiments use linear classifiers, which may underestimate the identity information available through nonlinear decoding.

**Broader impact and ethical considerations.** Cross-modal person matching is inherently dual-use: the same capability that enables legitimate applications such as multimedia indexing, accessibility tools, and forensic investigation can also facilitate surveillance, unauthorized tracking, or covert identification of individuals without their consent. Face and voice data are biometric identifiers that, once linked, create richer profiles than either modality alone, amplifying privacy risks. All datasets used in this work (MAV-Celeb and VoxCeleb1) are publicly available research datasets derived from media content where subjects appear in public contexts, and we use them solely for scientific evaluation of representation learning. Nevertheless, deployment of such systems in real-world settings raises concerns about consent, disproportionate impact on marginalized communities, and potential chilling effects on free expression. We encourage practitioners to adopt safeguards including informed consent, purpose limitation, access controls, and auditing mechanisms when building on this work. We further recommend that applications be evaluated for fairness across demographic groups, as biases in the underlying foundation models may propagate through cross-modal matching pipelines. Research in this area should be guided by institutional review and applicable data protection regulations (e.g., GDPR, BIPA) to ensure that advances in cross-modal biometric matching are developed and deployed responsibly.

# 6 Conclusion

We have shown that frozen unimodal foundation models, combined with lightweight projection heads, provide a simple and effective approach to cross-modal person matching. Our key finding is an informative asymmetry: Low-Rank Adaptation of the face encoder is highly effective (reducing EER from $19.9 \pm 0.3\%$ to $16.6 \pm 0.4\%$ on MAV-Celeb), while voice encoder adaptation provides no benefit. Layer-wise identity probing reveals that this asymmetry stems from a corresponding imbalance in pretrained identity encoding: WavLM already captures strong speaker identity (90–94% probe accuracy), while SigLIP's face identity encoding is substantially weaker (58–80%).

This finding replicates across three datasets (MAV-Celeb, VoxCeleb1, and CN-Celeb-AV), multiple evaluation protocols (verification, retrieval, $N$-way matching), and cross-dataset transfer experiments. Replacing SigLIP with DINOv2 - which achieves 96.8% face identity probing accuracy compared to SigLIP's 79.5% - confirms the prediction: LoRA provides no benefit when the encoder already encodes identity well. Our best model achieves $16.6 \pm 0.4\%$ EER on MAV-Celeb with only 1.33M trainable parameters.

More broadly, our results suggest that when bridging heterogeneous foundation models for cross-modal tasks, the effectiveness of adaptation is determined by the gap between each encoder's pretraining objective and the target task. In our setting, identifying and adapting the "weaker" encoder - rather than uniformly adapting all components - yields both better performance and better parameter efficiency. Testing whether this hypothesis extends to other encoder pairs and cross-modal tasks is an important direction for future work.

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

## A  Implementation Details

**Audio preprocessing.**  Raw waveforms are resampled to 16 kHz and truncated or zero-padded to 5 seconds (80,000 samples). No augmentation is applied; WavLM's feature extractor handles normalization internally. An attention mask excludes padding frames during mean pooling.

**Image preprocessing.**  Face images are resized to $384 \times 384$ pixels and preprocessed using SigLIP's standard image processor (normalization with ImageNet statistics). Face detection and cropping are performed as a preprocessing step using dataset-provided annotations (MAV-Celeb) or automated face detection (VoxCeleb1).

**LoRA configuration.**  LoRA adapters are injected into the $W_q$ and $W_v$ matrices of all transformer self-attention layers. For SigLIP ViT-B/16 (12 layers): $12 \times 2 \times 2 \times r \times d = 12 \times 2 \times 2 \times 4 \times 768 = 147{,}456$ parameters. For WavLM-Large (24 layers): $24 \times 2 \times 2 \times r \times d = 24 \times 2 \times 2 \times 4 \times 1024 = 393{,}216$ parameters. We use $\alpha = 2r = 8$ and no dropout.

**Evaluation protocol.**  For verification, we compute cosine similarity between all voice-face pairs in the test set and report EER as the point where false acceptance rate equals false rejection rate. For retrieval, we compute per-identity mean embeddings across all test samples, re-normalize to unit norm, and report the fraction of queries where the correct identity appears in the top-$K$ results, averaged across voice-to-face and face-to-voice directions.

**Compute.**  All experiments were conducted on NVIDIA GPUs (RTX 3090, 24GB) (RTX 5060Ti, 16GB). Projection-only models train in approximately 1–2 hours (30 epochs, MAV-Celeb) or 9 hours (15 epochs, VoxCeleb1). LoRA experiments require approximately 3–4 hours (MAV-Celeb) or 9 hours (VoxCeleb1) due to gradient computation through the adapted encoder layers.

## B  Probing Experiment Details

For each encoder layer, we extract representations for all samples in the dataset (up to 50 samples per identity for MAV-Celeb, up to 20 per identity for VoxCeleb1). WavLM representations are mean-pooled over the time dimension; SigLIP representations use the CLS token. We train an $\ell_2$-regularized logistic regression classifier (sklearn, $C = 1.0$, max 1000 iterations, `lbfgs` solver) with stratified 80/20 train/test splits.

Table 8 reports full layer-wise results.

## C  Training Dynamics

Table 9 shows the evolution of metrics during training for the projection-only model on MAV-Celeb. Performance improves rapidly in the first 10 epochs. EER plateaus around 19–20% beyond epoch 10, while Rank-1 continues to improve modestly, suggesting diminishing returns from extended training.

Table 8: **Full layer-wise probing results** (test accuracy). WavLM-Large has 25 layers (0 = CNN features, 1–24 = transformer layers). SigLIP ViT-B/16 has 13 layers (0 = patch embeddings, 1–12 = transformer layers).

| | MAV-Celeb (70 IDs) | | VoxCeleb1 (1,211 IDs) | |
|---|---|---|---|---|
| **Layer** | **WavLM** | **SigLIP** | **WavLM** | **SigLIP** |
| 0 | 77.2% | 19.9% | 51.3% | 16.4% |
| 1 | 85.6% | 52.7% | 62.5% | 39.5% |
| 2 | 89.9% | 65.1% | 73.0% | 50.0% |
| 3 | 92.4% | 68.0% | 81.8% | 53.0% |
| 4 | 92.8% | 71.1% | 82.5% | 53.9% |
| 5 | 93.1% | 75.3% | 86.1% | 55.5% |
| 6 | **93.8%** | 76.9% | 90.2% | 56.6% |
| 7 | 92.5% | 77.2% | 90.2% | 56.7% |
| 8 | 92.6% | **79.5%** | 89.9% | 56.9% |
| 9 | 92.5% | 78.4% | **90.5%** | 58.1% |
| 10 | 90.9% | 77.3% | 89.6% | **58.1%** |
| 11 | 90.6% | 74.9% | 88.1% | 56.9% |
| 12 | 89.2% | 69.4% | 84.3% | 55.0% |
| 13 | 87.3% | — | 80.7% | — |
| 14 | 83.8% | — | 75.3% | — |
| 15 | 82.7% | — | 69.9% | — |
| 16 | 82.1% | — | 65.6% | — |
| 17 | 80.2% | — | 60.5% | — |
| 18 | 77.5% | — | 56.2% | — |
| 19 | 76.0% | — | 54.1% | — |
| 20 | 77.6% | — | 54.0% | — |
| 21 | 79.9% | — | 62.0% | — |
| 22 | 79.1% | — | 63.2% | — |
| 23 | 77.2% | — | 59.4% | — |
| 24 | 62.8% | — | 35.0% | — |
| **Best** | **93.8%** (L6) | **79.5%** (L8) | **90.5%** (L9) | **58.1%** (L9–10) |
| **Gap** | 14.3 pp | | 32.4 pp | |

Table 9: Training progression for the projection-only model on MAV-Celeb v1.

| Epoch | Rank-1 | EER↓ | AUC↑ |
|---|---|---|---|
| 5 | 28.6% | 25.5% | 0.818 |
| 10 | 61.4% | 19.6% | 0.888 |
| 15 | 69.3% | 18.6% | 0.893 |
| 20 | 67.9% | 19.9% | 0.885 |
| 25 | 72.9% | 19.8% | 0.885 |
| 30 | 71.4% | 19.9% | 0.884 |

