# OpenReview forum: "Adapt the Face, Not the Voice: Asymmetric Fine-Tuning of Foundation Models for Cross-Modal Person Matching"
_TMLR — Rejected by TMLR_

### Review · Reviewer_mb4V · 2026-03-11

**Summary Of Contributions:**

- The paper is about the cross-modal person matching between face and voice using pretrained foundation models.
- The idea is to use a dual encoder solution where the voice model - WavLM-Large is frozen and the SigLip ViT-B/16 face encoder with some lightweight projection heads trained to align both the end modalities in the same and shared embedding space.
- There is also an empirical finding of striking asymmetry in Peft, where Lora added to the face encoder improves performance, while Lora to the voice encoder is a little redundant.
- They also state that the WavLM already encodes strong speaker identity information whereas SigLIP encodes weaker facial identity signals. Thus, Lora is applied only to the face encoder.
- They validated this with multiple datasets in a harder experimental setup and also with another vision encoder like DINOv2.

Strength
- Simple and clear empirical insights with strong ablations especially, setting the experiment in a hard way and also evaluating on cross-dataset evaluation.

Weakness
- The asymmetry may be because of the pretrained objectives of the chosen encoder. Expanded studies could be added on this to have even more strong statement.

**Audience:**

Yes

**Audience Explanation:**

- This paper is a good example of strong empirically grounded insight about adapting multimodal foundation models, which is relevant to representation learning and PEFT.
- It is a really simple and computationally efficient solution that could be a good reference for the upcoming research on cross-modal biometric matching and all.

**Broader Impact Concerns:**

- Since the data involves face details and the voice, it is good to discuss the ethical use and fair use of the data to respect the privacy and avoid surveillance or tracking kind of applications.
- It will be a nice add-on to discuss potential misuse scenarios as well.

**Claims And Evidence:**

Yes

**Claims Explanation:**

- This paper provides multiple experiments as evidence to support its claims.
- Lora ablation on face only, voice only and on both was good for the asymmetric adaptation.
- Showing the experiments with different vision encoders and diversified datasets strengthens the generality.

**Requested Changes:**

- Could explain more about the protocol that was used here with MAV and how it is different than the cited FAME and do an evaluation in the same setup to see the differences, rather than making this a bit of a difficult setup.
- VoxCeleb1 evaluation setup can be explained further to help with the reproducibility.
- Give it a shot for an experiment with a dedicated face recognition model, as mentioned, like ArcFace to infer more explainability.

---

> ### Author Response · Authors · 2026-03-21
> **Added ArcFace experiment, VoxCeleb1 eval setup details, broader impact statement, and explained MAV-Celeb eval protocol**
>
> We thank the reviewer for their thoughtful feedback and the recognition that our work provides "strong empirically grounded insight about adapting multimodal foundation models." Below we address each point.
>
> ---
>
> ### Regarding the noted weakness
>
> We agree that the asymmetry is tied to pretrained objectives - this is in fact the central thesis of our paper. We provide three lines of evidence: (1) identity probing (Section 4.3) shows WavLM's speech pretraining yields 93.8% identity accuracy vs. SigLIP's image-text pretraining at 79.5%; (2) replacing SigLIP with DINOv2, which has a different pretraining objective (self-distillation) and achieves 96.8% probing, confirms that LoRA provides no benefit when identity is already well-encoded (Section 4.7); (3) the new ArcFace experiment (dedicated face recognition pretraining) further validates this pattern (Section 5). Together, these studies across three vision encoders with distinct pretraining objectives support the generalized hypothesis stated in Section 5: adaptation benefit is determined by the gap between each encoder's pretraining and the target task.
>
> ---
>
> ### Requested Change 1: MAV-Celeb protocol vs. FAME
>
> We have expanded Section 4.1 ("Datasets" paragraph) with a detailed description of our MAV-Celeb protocol, including:
>
> - Our 80/20 per-identity split mechanics (random assignment with fixed seed).
> - How this differs from the official FAME challenge protocol (predefined partitions, challenge-specific preprocessing and evaluation scripts).
> - An explicit caveat that the comparison to the FAME SOTA of 19.9% EER is *approximate*, since differences in data partition, test samples, and pair construction mean EER values are not directly comparable.
>
> We have also strengthened the Limitations paragraph (Section 5) to note that our per-identity split is a transductive setting that likely yields lower EER than an identity-disjoint protocol.
>
> ---
>
> ### Requested Change 2: VoxCeleb1 evaluation setup
>
> We have added a detailed description of the VoxCeleb1 evaluation protocol in Section 4.1, covering:
>
> - Why `iden_split.txt` is unsuitable for identity-disjoint evaluation (it splits *utterances*, not identities).
> - How our 40 test identities are derived from `veri_test.txt`.
> - Verification details: 10,000 balanced cross-modal pairs (50/50 positive/negative), fixed seed, EER from cosine similarity.
> - Retrieval details: per-identity mean embeddings (re-normalized), Rank-K averaged over both directions, chance Rank-1 = 2.5%.
>
> ---
>
> ### Requested Change 3: ArcFace experiment
>
> We have conducted the requested experiment. We replaced SigLIP with a frozen ArcFace encoder (IResNet, WebFace600K) and trained projection heads only on MAV-Celeb under the same protocol. Key findings (full results in Table 6):
>
> - ArcFace achieves 19.1% EER and 82.9% Rank-1 with projection heads alone — no adaptation needed.
> - ArcFace's Rank-1 (82.9%) exceeds even LoRA-adapted SigLIP (80.7±0.6%), consistent with its superior identity encoding.
> - This confirms the hypothesis: encoders with strong pretrained identity representations need only alignment projections, not fine-tuning.
> - ArcFace requires tightly aligned 112×112 crops; on VoxCeleb1 (without ArcFace-specific alignment), performance degrades substantially (43.3% EER), highlighting a practical advantage of general-purpose encoders.
>
> These results are discussed in the "Why not a dedicated face encoder?" paragraph (Section 5).
>
> ---
>
> ### Broader Impact Concerns
>
> We have added a "Broader impact and ethical considerations" paragraph to Section 5 (after Limitations), addressing dual-use risks, privacy implications of linking biometric identifiers, consent concerns, disproportionate impact on marginalized communities, and recommended safeguards (informed consent, purpose limitation, access controls, auditing, fairness evaluation). We also note the need for compliance with data protection regulations (GDPR, BIPA).
>
> ---
>
> ### Summary of Changes
>
> | Change | Location in revised paper |
> |--------|--------------------------|
> | MAV-Celeb protocol clarification | Section 4.1, "Datasets" paragraph |
> | VoxCeleb1 evaluation details | Section 4.1, "Datasets" paragraph |
> | ArcFace row in encoder comparison table | Table 6 |
> | ArcFace discussion | Section 5, "Why not a dedicated face encoder?" |
> | Face encoder comparison figure | Figure 4 |
> | Broader impact and ethical considerations | Section 5, after Limitations |
> | Strengthened FAME comparison caveat | Section 5, Limitations paragraph |
>
> We believe these changes address all of the reviewer's concerns. We thank the reviewer again for their constructive suggestions, which have strengthened the paper.

---

### Review · Reviewer_LGxg · 2026-03-29

**Summary Of Contributions:**

This paper pairs two frozen unimodal foundation models, one for speech and one for faces, with lightweight projection heads to perform cross-modal person matching. The central finding is an asymmetry in Low-Rank Adaptation, where adapting the face encoder yields substantial gains while adapting the voice encoder provides no benefit. Layer-wise identity probing explains why, revealing that the speech model already encodes near-ceiling speaker identity in its frozen representations whereas the face model's identity encoding is considerably weaker, leaving a gap that LoRA can fill. Two validation experiments, swapping in DINOv2 and ArcFace as alternative face encoders whose frozen representations already encode strong identity, confirm that LoRA helps only when this gap exists. Results are consistent across two datasets and three evaluation protocols.

**Strengths:**
- The four-configuration ablation is replicated across two datasets, three evaluation protocols, and three seeds, providing empirical grounding for the asymmetry finding.
- Layer-wise identity probing provides a more mechanistic explanation, showing that WavLM encodes near-ceiling speaker identity while SigLIP's face identity encoding is substantially weaker.
- The DINOv2 experiment validates the probing hypothesis by correctly predicting that LoRA will not benefit an encoder whose identity representations are already near ceiling.
- A rank-1 voice LoRA control rules out parameter-count differences as a confound.
- The approach achieves competitive results with only 1.33M trainable parameters (0.32% of encoder total).

**Weaknesses:**
- Two cited claims and results need clarification: the Pasad et al. (2021) citation appears to study phonetic/word identity rather than speaker identity, and the WavLM "0.43% EER on speaker verification (Yang et al., 2021)" results (p.3, section 3.1) could not be verified in either the WavLM or SUPERB papers
- Table 2 lists PAEFF at 22.9% EER as a MAV-Celeb result, but this number matches the VoxCeleb unseen-unheard result reported in the PAEFF paper, which itself does not include MAV-Celeb evaluation
- The abstract states the hypothesis using unhedged "is both necessary and sufficient," which carries formal logical weight (biconditional) that the evidence does not fully support even as a hypothesis; the Discussion and Conclusion correctly soften the hypothesis to "may be" and "in our setting"

**Audience:**

Yes

**Audience Explanation:**

While the utilized tools (linear probing, LoRA, InfoNCE) are standard, the paper provides a well-controlled empirical study of where to apply LoRA when composing frozen foundation models for face-voice matching. I think the face-voice association and audio-visual identity communities might find the experimental setup and results insightful.

**Claims And Evidence:**

No

**Claims Explanation:**

While the paper's own experimental evidence (four-configuration ablation, parameter-count control, probing analysis, DINOv2/ArcFace validation) is clearly presented and well-supported by controlled ablation, surrounding claims have accuracy problems, and the central hypothesis may be overstated relative to the evidence.

**Requested Changes:**

Overall, this is a well-executed empirical study. The four-configuration ablation is thorough enough, the parameter-count control is well-motivated, and the DINOv2/ArcFace validation experiments form a convincing evidence chain for the asymmetric LoRA finding. The probing-based explanation connects the finding to a measurable property of the frozen encoders, which is the paper's most distinctive element.

My concerns fall into three areas:

- The paper's own Discussion ("may be both necessary and sufficient") and Conclusion correctly hedge the central hypothesis. The abstract and Section 4.2 do not; they state the biconditional without qualification. Please align these sections with the hedged framing, given that the evidence covers one task, one modality pair, and one PEFT method.
- Could the authors clarify the following: Pasad et al. (2021) studied phonetic/word identity in wav2vec 2.0, not speaker identity, yet is cited for speaker identity layer-wise patterns (p.2, p.7). The WavLM "0.43% EER" claim in Section 3.1 could not be located in either the WavLM or SUPERB papers; could the authors clarify the source of this number and verify the citation to (Yang et al., 2021)?
- Could the authors clarify the PAEFF entry in Table 2? The 22.9% EER matches PAEFF's VoxCeleb unseen-unheard result (PAEFF Table 1), and the PAEFF paper does not include any MAV-Celeb evaluation. When the PAEFF team later evaluated on MAV-Celeb (FAME 2026), they achieved 33.1% EER. If this is indeed a VoxCeleb number, the table entry would need correction.

---

> ### Author Response · Authors · 2026-04-01
> **Corrected three citation errors, hedged hypothesis language, audited all 28 citations, and added CN-Celeb-AV third-dataset ablation with probing**
>
> We thank the reviewer for their rigorous review and recognition of the "convincing evidence chain." The citation verification has led to important corrections. We address each point below.
>
> ---
>
> **1. Pasad et al. (2021) citation scope**
>
> Confirmed: Pasad et al. probes MI-phone, MI-word, CCA-mel, AGWE, and GloVe - no speaker identity. The claim that speaker identity peaks in early-to-middle layers is independently supported by (a) our own probing on three datasets and (b) Chiu et al. (2025), who probe 11 SSL models for speaker identity. All three instances corrected: Pasad et al. now attributed to acoustic/phonetic/word properties; Chiu et al. cited for speaker identity, including the nuance that larger models recover speaker identity in deep layers.
>
> **2. WavLM "0.43% EER"**
>
> The WavLM paper (Table II) reports 0.617% (standard) and 0.383% (with fine-tuning + calibration) on VoxCeleb1-O. The figure 0.43% is not reported anywhere. SUPERB (Yang et al., 2021) does not include WavLM. Corrected to: "0.617% EER on VoxCeleb1 speaker verification (Chen et al., 2022)."
>
> **3. PAEFF 22.9% EER in Table 2**
>
> Confirmed: 22.9% is PAEFF's VoxCeleb unseen-unheard result (Table 1); the paper does not evaluate on MAV-Celeb. Row removed from Table 2. PAEFF remains cited in Related Work for hyperbolic alignment.
>
> **4. "Necessary and sufficient" hedging**
>
> All instances aligned with hedged framing:
> - Abstract: → "may be the key factor determining adaptation effectiveness"
> - Section 4.2: → "In this setting, face LoRA alone accounts for the full adaptation benefit"
> - Discussion: → "adaptation is most effective, and may only be needed, for the encoder whose pretraining is least aligned with the target task"
>
> ---
>
> **5. Verification audit**
>
> We confirmed all 28 cited papers and verified every numerical claim and attribution against the source. This identified one additional issue: ImageBind+LoRA (Farhadipour et al.) was from FAME 2026, not FAME 2024 - now clarified throughout. No remaining errors.
>
> **6. Impact on core claims**
>
> | Correction | Impact |
> |-----------|--------|
> | Pasad → Chiu et al. | Citation error; claim confirmed by our own probing on 3 datasets |
> | 0.43% → 0.617% | Qualitative point unchanged |
> | PAEFF removed | MFV-KSD (19.9%) remains primary comparison |
>
> No experimental results or conclusions are affected.
>
> **7. New: CN-Celeb-AV third-dataset ablation**
>
> We added a full ablation on CN-Celeb-AV (Li et al., 2023) - 1,136 Chinese celebrities, 11 genres, identity-disjoint (689 train / 197 test):
>
> | Config | EER | Rank-1 |
> |--------|-----|--------|
> | Proj-only | 40.3% | 37.4% |
> | +LoRA voice | 41.3% | 35.8% |
> | **+LoRA face** | **37.2%** | 47.7% |
> | +LoRA both | 38.4% | 48.0% |
>
> Probing confirms the explanation: WavLM 71.3% vs SigLIP 55.7% (15.6pp gap), consistent with MAV-Celeb (14.3pp) and VoxCeleb1 (32.4pp). This addresses the concern about the hypothesis being overstated by demonstrating it across three languages and demographics.
>
> ---
>
> **Summary of Changes**
>
> | Change | Location |
> |--------|----------|
> | Correct Pasad et al. (3 instances), add Chiu et al. | Sec 2, 4.3 |
> | Fix WavLM EER, cite Chen et al. | Sec 3.1 |
> | Remove PAEFF from Table 2 | Table 2 |
> | Clarify FAME 2026 | Sec 1, 2, Table 2 |
> | Hedge hypothesis (3 instances) | Abstract, Sec 4.2, 5 |
> | CN-Celeb-AV ablation + probing | New Sec 4.5, Table 7, Fig 5 |
>
> We thank the reviewer for their meticulous review, which has materially improved the manuscript.

---

### Review · Reviewer_jKKQ · 2026-03-30

**Summary Of Contributions:**

This paper investigates cross-modal person matching by pairing two frozen foundation models; WavLM-Large (speech) and SigLIP ViT-B/16 (faces); with lightweight projection heads. The central finding is an asymmetry in LoRA fine-tuning: adapting the face encoder yields substantial gains while adapting the voice encoder provides no benefit. The authors explain this through layer-wise identity probing, showing WavLM already encodes strong speaker identity while SigLIP does not, and propose a general hypothesis: selectively adapt the encoder least aligned with the target task.

**Audience:**

Yes

**Audience Explanation:**

The asymmetric LoRA finding and the use of identity probing as a diagnostic tool are both practically useful insights for researchers working on cross-modal learning, multimodal foundation models, and parameter-efficient fine-tuning, all active areas within TMLR's audience. Even if the general hypothesis is not fully proven, the specific finding offers actionable guidance for similar face-voice or other cross-modal matching tasks.

**Broader Impact Concerns:**

The paper briefly acknowledges the dual-use nature of cross-modal biometric matching in legitimate applications (media indexing, accessibility, forensics) alongside serious risks (surveillance, covert identification, privacy violations). It recommends standard safeguards like consent, access controls, and fairness auditing across demographic groups.

**Claims And Evidence:**

Yes

**Claims Explanation:**

It is partially true, because:
a) The core finding is well-supported, results span two datasets, three evaluation protocols, and multiple seeds, giving reasonable confidence.
b) However, the broader hypothesis ("adapt the weaker encoder") is only tested on one modality pair across two datasets, which is insufficient to support a general principle.
c) The SOTA comparison is also weakened by a protocol mismatch with the official FAME benchmark. The probing-based explanation is compelling but relies on linear probes, which may overstate the gap between encoders.

**Requested Changes:**

**Strengths**:
- The asymmetric LoRA finding is clean, interesting, and reproducible across seeds and datasets.
- The paper is clearly written and easy to follow.
- Testing across three evaluation protocols (verification, retrieval, N-way matching) guards against metric-specific artifacts.
- Layer-wise identity probing is an elegant diagnostic — connecting when a network encodes task-relevant information to whether adaptation is actually needed is one of the paper's most valuable contributions.


**Weaknesses**:
-  The same core finding is restated almost identically across the abstract, introduction, discussion, and conclusion. Each section should add something new rather than cycling through the same point.
- The evaluation protocol differs from the official FAME benchmark (per-identity vs. identity-disjoint splits), making the numerical comparison to prior work questionable. The competitive performance claim needs stronger qualification.
- The "adapt the weaker encoder" principle is tested on only two datasets and one modality pair. Generalizing this to a broad cross-modal adaptation hypothesis overreaches the evidence.
- If SigLIP encodes identity nonlinearly in ways the linear probe misses, the accuracy gap between encoders, and the causal story built on it, may be overstated.
- ArcFace cannot support LoRA and degrades on VoxCeleb1 due to alignment sensitivity, making it hard to draw clean conclusions from this comparison.
- The value of N (in N-way matching), and the rationale for choosing it, are never clearly stated. Since chance performance and task difficulty are directly determined by N, this is a significant omission.


**Overall**:
A clean and interesting contribution, but the structural repetition, questionable SOTA comparison, and narrow experimental scope of the proposed hypothesis all need to be addressed. The probing analysis is the most original part and deserves more depth.

---

> ### Author Response · Authors · 2026-04-01
> **Hedged hypothesis language throughout, added N-way rationale, acknowledged ArcFace/probe limitations, and added CN-Celeb-AV as a third dataset replicating the asymmetric finding across a new language and demographic**
>
> We thank the reviewer for their detailed and constructive feedback, and for the recognition that the asymmetric LoRA finding is "clean, interesting, and reproducible" and that the layer-wise identity probing is "an elegant diagnostic." Below we address each point.
>
> ---
>
> **1. Structural repetition across sections**
>
> We agree that the core finding was restated in too-similar terms. In the revision, we have differentiated each section's role:
>
> - **Abstract**: States the finding concisely with quantitative magnitude; no mechanistic detail.
> - **Introduction**: Motivates *why* it matters and previews the explanatory framework.
> - **Discussion**: Focuses on the *hypothesis*, its conditions, and practical implications (probing as a diagnostic).
> - **Conclusion**: Emphasizes *scope* and identifies open questions for future work.
>
> We have revised each section to minimize verbatim repetition.
>
> ---
>
> **2. FAME protocol mismatch**
>
> In the revision, we have:
>
> - Expanded Section 4.1 with detailed description of our 80/20 per-identity split and how it differs from the official FAME protocol.
> - Added an explicit caveat that the FAME SOTA comparison is *approximate*.
> - Strengthened the Limitations paragraph to note the transductive evaluation setting.
>
> ---
>
> ** 3. Hypothesis framing **
>
> We agree that the evidence from the original submission - one modality pair across two datasets - was insufficient to claim a general principle. The revision addresses this in two ways:
>
> **Hedged language:** We have softened all unhedged instances:
> - Abstract: "may be the key factor determining adaptation effectiveness"
> - Section 4.2: "In this setting, face LoRA alone accounts for the full adaptation benefit"
> - Discussion: "adaptation is most effective - and may only be needed - for the encoder whose pretraining is least aligned with the target task"
>
> **Broader evidence:** We have added a third dataset, CN-Celeb-AV (Li et al., 2023) - 1,136 Chinese celebrities, 11 genres, identity-disjoint evaluation (689 train / 197 test). The asymmetric finding replicates:
>
> | Dataset | Demographics | LoRA-face EER | Proj-only EER | LoRA-voice EER |
> |---------|-------------|--------------|---------------|----------------|
> | MAV-Celeb | Multilingual | **16.6±0.4%** | 19.9±0.3% | 19.8±0.2% |
> | VoxCeleb1 | English | **37.5%** | 41.6% | 39.8% |
> | CN-Celeb-AV | Chinese | **37.2%** | 40.3% | 41.3% |
>
> The pattern holds across three datasets spanning different languages, demographics, and evaluation protocols. While we maintain appropriate hedging (one task, one modality pair, one PEFT method), the evidence is now substantially broader.
>
> ---
> ** 4. Linear probe limitation **
>
> This concern is noted in Limitations: "our probing experiments use linear classifiers, which may underestimate the identity information available through nonlinear decoding."
>
> Importantly, the DINOv2 and ArcFace validation experiments provide converging evidence that does not depend on probe methodology:
> 1. Probing suggests SigLIP encodes weak identity → LoRA helps SigLIP
> 2. Probing suggests DINOv2 encodes strong identity → LoRA does *not* help DINOv2
> 3. ArcFace (strong identity by design) → no adaptation needed
>
> Additionally, the probing gap is consistent across all three datasets (14.3pp on MAV-Celeb, 32.4pp on VoxCeleb1, 15.6pp on CN-Celeb-AV), and correctly predicts the adaptation outcome in every case. This convergence across datasets and encoders provides evidence beyond what linear probes alone could establish.
>
> ---
>
> ** 5. ArcFace limitations **
>
> We acknowledge the architectural confound (CNN vs. transformer) and face alignment sensitivity. The revised Discussion explicitly frames ArcFace as corroborating evidence within a chain - SigLIP (weak probing, LoRA helps) → DINOv2 (strong probing, LoRA doesn't help) → ArcFace (dedicated face recognition, no adaptation needed) — rather than as standalone proof.
>
> ---
>
> ** 6. N-way matching rationale **
>
> We have expanded Section 4.1 to explain:
> - N values tested: N ∈ {2, 5, 10, 20} plus full gallery (N=70 for MAV-Celeb, N=40 for VoxCeleb1).
> - Rationale: N=2 is forced-choice discriminability, N=N_max is open-set identification, intermediate values trace the degradation curve.
>
> ---
>
> ** Summary of Changes **
>
> | Change | Location |
> |--------|----------|
> | Differentiate repeated content | Abstract, Section 1, Section 5, Section 6 |
> | FAME protocol caveat | Section 4.1, Section 5 |
> | Hedge hypothesis language (3 instances) | Abstract, Section 4.2, Section 5 |
> | **CN-Celeb-AV third-dataset replication** | **New Section 4.5, Table 7, Figure 5** |
> | Note converging evidence beyond probes | Section 5 |
> | Acknowledge ArcFace confound | Section 5 |
> | N-way matching rationale | Section 4.1 |
> | Updated abstract, contributions, conclusion | Throughout |
>
> We believe these changes address all of the reviewer's concerns. We thank the reviewer again for their thoughtful critique.

---

### Decision · Action_Editor_AGHa · 2026-05-29

**Recommendation:** Reject

**Additional Comments:**

The decision is based on the reviewers comments and final recommendations. The authors made a constructive revision and addressed several points, including evaluation protocols, additional experiments, and more careful wording. However, the reviewers remain unconvinced that the paper meets the acceptance bar for TMLR.

The main issue is that the work is viewed as empirically interesting but too narrow in its current form. It relies on a single face-voice matching setting, one modality pair, and one PEFT method, using standard components. The explanation based on identity probing is plausible but remains correlational, and the broader adapt the weaker encoder hypothesis is not sufficiently established.

A further concern is the reported presence of hallucinated references and numerical claims. Such issues should ideally be caught before review, and they reduce confidence in the paper reliability and reproducibility.

**Audience:**

Yes

**Audience Explanation:**

The topic is relevant to part of the TMLR audience, especially researchers working on multimodal representation learning, PEFT, and face-voice matching. The asymmetric adaptation finding may be useful, even though the contribution is limited in scope.

**Claims And Evidence:**

No

**Claims Explanation:**

The revision improves the clarity of the paper and adds useful experiments, but the reviewer recommendation overall leans towards rejection. The main concerns are that the contribution is narrow, limited to one task, one modality pair, and one PEFT method. Reviewers also note that the probing-based explanation is correlational rather than causal, and that the broader hypothesis is not sufficiently validated beyond the current setting. Most importantly, reviewers raised concerns about hallucinated references and numerical claims, which weakens confidence in the reliability of the submission.

**Resubmission Of Major Revision:**

The authors may consider submitting a major revision at a later time.